# Use of Next Generation Sequencing to Define the Origin of Primary Myelofibrosis

**DOI:** 10.3390/cancers15061785

**Published:** 2023-03-15

**Authors:** Giuseppe Visani, Maryam Etebari, Fabio Fuligni, Antonio Di Guardo, Alessandro Isidori, Federica Loscocco, Stefania Paolini, Mohsen Navari, Pier Paolo Piccaluga

**Affiliations:** 1Hematology and Stem Cell Transplantation, AORMIN, 61121 Pesaro, Italy; 2Department of Medical Biotechnology, School of Paramedical Sciences, Torbat Heydariyeh University of Medical Sciences, Torbat Heydariyeh 33787-95196, Iran; 3Research Center of Advanced Technologies in Medicine, Torbat Heydariyeh University of Medical Sciences, Torbat Heydariyeh 33787-95196, Iran; 4Department of Medical Science and Surgery (DIMEC), University of Bologna, 40126 Bologna, Italy; 5The Hospital for Sick Children, Toronto, ON M5G 1X8, Canada; 6Biobank of Research, IRCCS Azienda Ospedaliero-Universitaria di Bologna, 40138 Bologna, Italy; 7Bioinformatics Research Center, Mashhad University of Medical Sciences, Mashhad 91778-99191, Iran

**Keywords:** primary myelofibrosis, next generation sequencing, cell sorting, single nucleotide variants, cell of origin, whole exome sequencing

## Abstract

**Simple Summary:**

The cell of origin (a stem cell) of primary myelofibrosis (PMF) is still partially unknown. We used next generation sequencing to explore this aspect of PMF pathobiology deeper. We found evidence that higher progenitors (i.e., pluripotent stem cells) harbor genetic lesions found in PMF cells. However, genetic events can also occur in later stages of differential contribution to disease development and progression. This new finding sheds light on the complexity of the transformation of hematopoietic pluripotent precursors in the genesis of PMF.

**Abstract:**

Primary myelofibrosis (PMF) is a chronic myeloproliferative neoplasm (MPN) characterized by progressive bone marrow sclerosis, extra-medullary hematopoiesis, and possible transformation to acute leukemia. In the last decade, the molecular pathogenesis of the disease has been largely uncovered. Particularly, genetic and genomic studies have provided evidence of deregulated oncogenes in PMF as well as in other MPNs. However, the mechanisms through which transformation to either the myeloid or lymphoid blastic phase remain obscure. Particularly, it is still debated whether the disease has origins in a multi-potent hematopoietic stem cells or instead in a commissioned myeloid progenitor. In this study, we aimed to shed light upon this issue by using next generation sequencing (NGS) to study both myeloid and lymphoid cells as well as matched non-neoplastic DNA of PMF patients. Whole exome sequencing revealed that most somatic mutations were the same between myeloid and lymphoid cells, such findings being confirmed by Sanger sequencing. Particularly, we found 126/146 SNVs to be the e same (including *JAK2V617F*), indicating that most genetic events likely to contribute to disease pathogenesis occurred in a non-commissioned precursor. In contrast, only 9/27 InDels were similar, suggesting that this type of lesion contributed instead to disease progression, occurring at more differentiated stages, or maybe just represented “passenger” lesions, not contributing at all to disease pathogenesis. In conclusion, we showed for the first time that genetic lesions characteristic of PMF occur at an early stage of hematopoietic stem cell differentiation, this being in line with the possible transformation of the disease in either myeloid or lymphoid acute leukemia.

## 1. Introduction

The classic Philadelphia chromosome-negative myeloproliferative neoplasms (MPNs) are a group of chronic hematological diseases which include three major entities: polycythemia vera (PV), essential thrombocythemia (ET), and primary myelofibrosis (PMF), all of which are characterized by the clonal expansion of one or several myeloid cell lineages (including granulocytes, monocytic, endothelial-like, erythroid, and megakaryocytes cells) and bone marrow fibrosis [1]; however, these entities represent distinct pathogenic phenotypes [1]. Due to the clonal nature of MPNs, patients might experience transformation to another entity, or develop some types of leukemia, and rarely proceed to lymphoid malignancies [2,3,4,5,6,7,8,9,10,11]. Among the MPN entities, PMF (formerly known chronic idiopathic myelofibrosis) is the most complex and heterogeneous MPN entity both in terms of its clinical and biological characteristics and the distinctive criteria which consists of megakaryocyte hyperproliferation and bone marrow fibrosis [12,13].

PMF is characterized by a stepwise evolution. There is an initial prefibrotic/early stage, distinguished by hypercellular bone marrow. Subsequently, an overt fibrotic stage with marked fibrosis in the bone marrow occurs. The fibrotic stage of PMF is clinically characterized by leukoerythroblastosis in the blood, hepatomegaly, and splenomegaly [1,12,13].

Genetic studies have revealed that mutations in the *JAK2* gene (more typically *JAK2*V617F) are the most frequent (around 50–60%) genetic aberrations in PMF patients, which lead to the constitutive activation of the JAK/STAT pathway. However, patients lacking this mutation often have alternative mutations [14,15]. *CALR* mutations are found in about one fourth of PMF cases, and *MPL* mutations are found in less than 10% of cases. MPL and CALR also activate the JAK/STAT pathway [13,16]. In total, 10–15% of PMF cases are triple-negative for mutations in *JAK2*, *CALR*, and *MPL* [13,16]. Although these mutations are useful in distinguishing PMF from inflammatory conditions, they are not specific to PMF; mutations in these genes are found in essential thrombocythemia and polycythemia vera [17,18,19].

The application of next generation sequencing (NGS) later allowed the further definition of the genetic landscape of MPN. Garcia-Gisbert et al. detected molecular alterations on cell-free DNA (cfDNA) and paired granulocyte DNA in plasma samples of a series of patients with MPNs. cfDNA and granulocyte DNA showed an equivalent mutational profile. The 14 PMF patients, included in the study cohort, exhibited mutations in driver genes such as *JAK2, MPL*, and *CALR*. Non-driver mutations, identified in these patients, involved *TET2*, *ASXL1*, *SRSF2*, *IDH2*, *SF3B1*, and *EZH2* [20]. 

The cell of origin of PMF in the hematological cell development hierarchy has been a matter of contradiction. Some evidence, such as clonal myelopoiesis and the detection of *JAK2V617F* mutation in both myeloid and lymphoid cells in myeloproliferative neoplasms, has resulted in the hypothesis that PMF originates from a multipotent hematopoietic stem/progenitor cell (HSPC) [15,21]. In this regard, a high frequency of multipotent CD34+ stem cells has been identified in the peripheral blood of PMF patients [22,23]. Another study, however, suggested that CD133+/34+ stem cells could be the neoplastic origin cell, a conclusion derived from PMF development in vivo [24]. 

The most comprehensive study aiming to define the clonal evolution of myeloproliferative neoplasms on a molecular level was presented by Levine and colleagues [25]. In this study, the authors performed a single cell mutational profiling of 123 patients and found that a small number of clones frequently harbor co-occurring mutations in epigenetic regulators, while mutations in signaling genes often occur in distinct subclones, consistent with increasing clonal diversity [25]. They also assessed how targeted therapies could perturbate clone hierarchy and how immunophenotypes vary across genetically different clones [25]. 

In this study, we aimed to further investigate the cell of origin of PMF. To address this issue, we applied whole exome sequencing to both myeloid and lymphoid cells from PMF patients and confirmed the sparsely available evidence concerning the derivation of PMF from multipotent hematopoietic stem cells. 

## 2. Materials and Methods

### 2.1. Case Selection

Three PMF samples were collected at the Hematology and Stem Transplantation Unit, Azienda Ospedaliera Marche Nord, Pesaro, Italy. The pathological diagnosis was confirmed by at least 2 expert hematopathologists by a bone marrow trephine biopsy. The final diagnosis was then established based on clinical and pathological features. An additional amount of nine cases was included in the validation cohort studied by Sanger sequencing. The details on these patients had been previously reported [26]. A summary of their main clinical characteristics is reported in Appendix A. No case selection was applied.

Peripheral blood cells and saliva were collected, serving as neoplastic and non-neoplastic cells, respectively (see below), before treatment initiation. A detailed description of the three patients studied by WES is presented in Table 1.

### 2.2. Isolation and Purification of Myeloid and Lymphoid Cells from Peripheral Blood

In order to obtain granulocytes (as the myeloid cells) and lymphocytes, we followed a procedure that has been previously described [26]. First, we used Ficoll-Paque to perform a density gradient separation as instructed by the manufacturer (GE Healthcare, Madison, WI, USA). The pellet containing granulocytes was further subjected to a mild osmotic shock, in order to eliminate erythrocytes, and washed twice with PBS. To ensure the collection of adequate neoplastic components, a morphological evaluation of the granulocytic fractions was performed. In order to isolate lymphocytes, the layer containing PBMCs was further subjected to isolation by means of immunomagnetic labeling of CD3+ cells followed by a separation process using the MACS device (Milteniy Biotech, Bergisch Gladbach, Germany). A flow cytometric analysis was then employed to confirm the purity of the isolated CD3+ cells, as described before [26].

### 2.3. Whole Exome Sequencing

Total DNA was extracted from 9 samples (3 myeloid cells, 3 lymphocytes, and 3 saliva, taken from 3 patients, respectively) with QIAamp DNA mini kit Qiagen according to the manufacturer’s instructions (Qiagen, Milan, Italy). A Covaris instrument was used to shear one microgram of DNA into 100–500 bp fragments, following fragmentation quality control using a DNA-7500 kit (Agilent, Santa Clara, CA, USA). Further, based on Illumina’s TruSeq DNA Sample Preparation v2 Guide, DNA libraries were pre-enriched. Briefly, after end repair, specific adapters (adenylate 3’ends and ligate) were used in a PCR reaction to selectively enrich the DNA fragments with adaptor molecules on both ends. After purifying the product of the PCR libraries using AmpureXP beads (BeckmanCoulter, Brea, CA, USA), we implanted Illumina’s TruSeq Exome Enrichment Guide (Illumina, San Diego, CA, USA) to perform exome enrichment. Two hybridizations of 20 h using biotinylated bait-based were performed, and each was followed with a step of streptavidin magnetic bead binding, the product of which was eluted after washing. After the second elution, a PCR reaction of 10 cycles was performed for enrichment, followed by a quality control analysis of the enriched libraries using a DNA-1000 kit (Agilent, USA). Quant-it PicoGreen dsDNA Assay Kit was used for the quantification, according to the manufacturer’s protocol (Invitrogen, Life Technologies, Waltham, MA, USA). 

Using Illumina HiScan SQ (Illumina, San Diego, CA, USA), we sequenced the paired-end libraries (2 × 100 base pair), following the manufacturer’s instructions. On average, about 17 million 100 bp paired-ends raw reads were generated, and the theoretical coverage was calculated based on the hg19 RefSeq non-redundant exome length, which ranged from 21X to 60X. The FastQC V0.10.0 tool (http://www.bioinformatics.babraham.ac.uk/projects/fastqc/ (accessed on 8 January 2014)) was used for quality control, and the reads were mapped to Human GRCh37 Genome Assembly using Burrows-Wheeler Aligner version 0.6.1 [27]. In order to remove potential PCR duplicates, we used samtools command rmdup to detect and collapse multiple mapped read pairs with identical external coordinates [27]. Mapping quality score recalibration and local realignment around insertions and deletions (InDels) was performed using Genome Analysis Toolkit (GATK) [28]. Single-nucleotide variants (SNVs) and small insertions and deletions (INDELS) were identified separately using GATK Unified-Genotyper.

All the mutations detected were filtered using thresholds based on the quality, coverage, and strand of the mapped reads and according to the variants already present in public databases (Hapmap, dbSNP and 1000 genome project (GRCh38.p13) (http://www.ensembl.org/Homo_sapiens/Info/Index (accessed on 8 January 2014)). To reduce the presence of germline mutations, variants already present in saliva samples were excluded. Saliva was macroscopically not contaminated by blood. However, in order to exclude the possible puzzling effects of scattered contaminating blood cells, SNVs/InDels with allele frequencies below 10% were excluded.

The Annovar tool with the 20 May 2013 update (http://www.openbioinformatics.org/annovar/ (accessed on 8 January 2014)) was used for functional annotation of variants, including exonic functions and aminoacid changes and only non-synonymous variants, including stop-gain SNVs, splicings, and frameshift InDels were selected for further analysis.

All the mutations found were manually checked and explored using Integrative Genomic Viewer 2.03 [29].

Furthermore, we used Molecular Signature Database (MSigDB, http://software.broadinstitute.org/gsea/msigdb (accessed on 8 January 2014)) for the analysis of the gene ontology (GO) molecular function of the genes with novel SNVs. For the investigation of the molecular interaction between the mutated genes and the possible drug candidates, and thus for defining actionable genes, the Cognoscente database was utilized (http://vanburenlab.tamhsc.edu/cognoscente.html (accessed on 8 January 2014)) [30].

### 2.4. Validation Analysis Using Sanger Sequencing

To confirm the efficiency of WES, we performed Sanger sequencing on a selected array of the genes found to be mutated. For this aim, DNA was extracted from the granulocytes and lymphocytes using QIAamp DNA mini kit Qiagen (Milan, Italy) according to the manufacturer’s procedure (Qiagen, Italy) [31]. Primers for the candidate loci were designed using Primer 3.0. PCR amplification was performed, and the products were purified using the QIAquick PCR purification kit from Qiagen according to the manufacturer’s instructions [31]. The purified PCR products were then sequenced on an ABI3730 genetic analyzer. The Sanger sequencing results were compared with the WES results.

## 3. Results

### 3.1. Whole Exome Sequencing Revealed Novel as well as Previously Known SNVs and InDels

By means of exome capture and high-throughput sequencing, we analyzed granulocytes and lymphocytes in three PMF patients and filtered the results with non-malignant cells obtained from saliva. The number of mate pair reads for patients 1, 2, and 3 was 16 M, 8 M and 8 M, respectively, with more than 97% of total mapped reads in all cases. The mean coverage for the patients was 27X, 14X, and 13.3X, respectively, with more than 86% of the exomes covering at least 1X (Appendix A). 

The discovered genetic aberrations included both SNVs and InDels, (Figure 1 and Figure 2). In total, we discovered as many as 3000 non-synonymous SNVs in all the patients, comprising 1620 (patient 1), 897 (patient 2), and 629 (patient 3), among which 80 (5.19%), 40 (4.67%), and 26 (4.31%) included novel non-synonymous SNVs (not present in DBSNP and 1000 genomes) for the three patients, respectively (Figure 1A, Appendix A). Among the novel SNVs, we found 142 missense mutations and 4 stop-gains in all the patients (Figure 1B). In terms of mutation frequency, we observed the highest values for chromosome 1, followed by chromosome 19 (Appendix A). 

Among the mutated genes, several genes with notable roles in cancer emerged, including several members of the zinc finger family of transcription factors [32] and other molecules related to transcription regulation such as interferon gamma inducible protein 16 (*IFI16*) [33], important receptors such as erb-b2 receptor tyrosine kinase 4 (*ERBB4*) and fibroblast growth factor receptor 2 (*FGFR2*) [34,35], and genes regulating ubiquitination such as autocrine motility factor receptor (*AMFR*) and ubiquitin-like modifier activating enzyme 2 (*UBA2*) [36,37]. 

Additionally, we discovered a total number of 101 InDels in all patients, which included 14 (27.45%, patient 1), 6 (20.69%, patient 2) and 7 (33.3%, patients) novel frameshift InDels (Figure 2A). In total, the novel InDels consisted of 14 insertions and 13 deletions, all of which occurred in one patient except for three InDels in three distinct genes which occurred in two patients: *HLA-B, VSIG10L*, and *ZNF717* (Figure 2B,C). The chromosomal distribution of InDels did not reveal remarkable differences across chromosomes; we simply observed higher frequencies in chromosome 12 (4 occurrences) followed by chromosomes 3 and 19 (3 occurrences) (Table 2 and Appendix A).

Details on novel SNVs are reported in Appendix A.

### 3.2. Comparison of Mutational Status of Lymphocytes and Granulocytes Suggests the Transformation to Occur before Lineage Division

In order to clarify the probable stage(s) during hematopoiesis in which the mutations occurred, we compared the discovered mutations in granulocytes and lymphocytes. Very interestingly, this comparison revealed that a major part of the SNVs were common between the two cell lineages. In fact, 126 out of the 136 (93%) SNVs in each cell type were found to be common between them (Figure 3A). In the case of InDels, although the shared number was lower, almost half of them were the same (9 out of 19 in granulocytes and 9 out of 18 in lymphocytes) (Figure 3B). 

Furthermore, to gain insights into the biological functions of mutated genes, we analyzed the GO function of the genes found to be mutated both in granulocytes and lymphocytes. The enriched functions turned out to be related to important signaling functions such as those related to free nucleotides and kinase activity, plus transcription-related functions such as RNA plolymerase II (Appendix A). To identify potentially actionable genes, a further analysis of the above-mentioned genes was carried out with Cognoscente, revealing JAK2, FGFR2, and PTGIS as the main candidate drug targets for PMF, which were connected to therapies such as ruxolitinib, tofacitinib, palifermin, ponatinib and epoprostenol (Appendix A). 

### 3.3. Sanger Sequencing Fully Supported WES Results

In order to validate the efficiency of the WES, we used Sanger sequencing on some selected genes including *ADAMTSL3*, *TET1*, *EFCAB4B*, *KIR2DL1*, *IKBKE*, *TMEM216*, and *JAK2*. These genes were chosen based on their biological relevance and known role in cancer. The studied cases included the ones used for the WES plus nine additional ones (Appendix A). We could confirm the selected discovered mutations in the initial samples and found that some of them were recurrently mutated in the additional cohort, with a high frequency. Particularly, we found *TET1* to be affected in 10/12 cases, *KIR2DL1* in 9/12 cases, *TMEM216* in 7/12 cases, *JAK2* in 4/12 cases, while *EFCAB4B* and *IKBKE* were affected in in 2/12.

## 4. Discussion

Myeloproliferative neoplasms are clonal disorders and originate from hematopoietic stem/progenitor cells. Primary myelofibrosis, as a heterogeneous MPN entity, is characterized by the aberrant production of megakaryocytes and bone marrow fibrosis [12]. Some studies suggest a stem/progenitor cell as the cell of origin for this neoplasm [22,23,24,25]; however, where such a cell would be located during the hematological hierarchy development remains unclear, mainly in terms of the myeloid/lymphoid progenitor cell. The study of the mutational landscape of the tumor could provide new evidence for such a hypothesis, and several studies have followed such an approach by studying myeloid cells [38,39,40]. In a recent study, Miles et al. used a robust approach which included a mutational analysis by the genome sequencing of CD34+ cells from more than 100 MPN and AML patients [25]. In this study, some of the MPN patients were analyzed both in the MPN and AML phase, and the study included subjects with clonal hematopoiesis but no MPN/AML as the control. The authors could define the hierarchy of somatic mutation along the development and evolution of myeloid neoplasms [25]. In our study, we tried to further investigate the nature of PMF by performing whole exome sequencing on granulocytes (as a representative of the myeloid lineage) and lymphocytes as two different hematopoietic cell lineages in PMF patients.

First of all, we found a considerable number of SNVs and InDels in our samples, many of which were novel. Furthermore, comparing the mutational status of myeloid and lymphoid cells presented that a considerable number of SNVs, and to a lesser extent of InDels, was common among the two. This finding provides further evidence that such mutations could have occurred in the upstream of the committed myeloid and lymphoid progenitor level, suggesting a multipotent progenitor cell as the origin cell for this disease. The evidence that InDels, rather than SNVs, were largely distinct in myeloid and lymphoid components, may suggest their later occurrence and possible role in disease progression rather than initiation. Overall, we could confirm Miles’ results to the extent of our (more limited) investigation.

The possible multipotent cell of origin of PMF has already been suggested. A study by Xu et al. evaluated the presence of a high frequency of CD34+ HSPC in the peripheral blood of PMF patients and suggested that PMF could arise from multipotent CD34+ HSPC [22]. In addition, Triviai et al. suggested that such cells of origin for PMF, in addition to CD34, express CD 133 as well [24]. This was confirmed both in terms of cell frequency, as well as the potential of such cells to develop a tumor in xenograft mice models. Very interestingly, however, they observed no lymphoid cells in the model mice [24]. Although such results seem to contrast our findings, it should be noted that they were obtained in an in vivo mice model and might not reflect the natural course of the disease. 

Farina et al. also identified molecular alterations in CD34 + HSPCs in PMF patients by next-generation sequencing. These findings appear to confirm the origin of PMF as multipotent CD34+ HPSCs. The most frequent mutations concern driver genes such as *JAK2* and non-driver genes such as *ASXL1*, *IDH1/2*, *SRSF2*, *EZH1* [41]. Further studies confirm a similar mutational profile in circulating free DNA (cfDNA) and in paired granulocyte DNA obtained from the plasma of PMF patients [20].

One of the most common mutations reported in PMF patients is *JAK2V617F*. Very importantly, a study reported the carriage of this mutation in lymphoid cells as well, which further confirms our hypothesis regarding the cell of origin for of [40]. Indeed, we found this mutation in one of the samples, which might seem relatively low as the rate of this mutation in PMF is 50–60%, which could be explained by the low sample number in our study. Similarly, despite the fact that no specific case selection was applied, neither *CALR* nor *MPL* mutations were observed in the initial cohort of three patients. This makes our series quite unique as triple negative cases are not common. In this regard, it should be noted how the evolution of liquid biopsy and liquid biopsy-based biomarkers (cell-free DNA, extracellular vesicles, microparticles, circulating endothelial cells, etc.), might turn out to be useful in PMF and especially in triple negative cases to differentiate subtypes and secondary causes of erythrocytosis, thrombocytosis, and myelofibrosis, as well as to predict the development of thrombosis [42]. 

Since PMF shows a pure myeloproliferative phenotype, the risk of myeloid transformation in this disease is significantly high; however, PMF also shows a rare association with lymphoid transformation, which happens for CML [6,43,44,45,46,47]. Indeed, there are also studies which have represented independent clonal involvement in concurrent or progressed diseases [48,49]. In our study, in addition to the high number of common mutations between myeloid and lymphoid lineages, we found some mutations which seemed to have been developed independently in each cell type, i.e., they could have occurred at a later stage during cell lineage differentiation. Among the genes mutated in lymphocytes, there was *MED25*, *TAOK2*, *RERE*, *ZNF84*, and *NLRP6* which play significant roles in cell signaling, chromatin modification, and apoptosis. These mutations might explain the lymphoblastic leukemia transformation in PMF, which is associated with the advent of new clonal expansions in the disease. 

The main limitations of our analysis were represented by the small sample size and the fact that the study design was simpler than that which was presented by Levine and colleagues [25]. Of note, however, is the fact that we could draw similar (though less extensive) conclusions. In addition, we should acknowledge that the use of saliva as a control is not always optimal as blood contamination may happen. However, in this specific case, the consistent differences between pathological samples (myeloid and lymphoid) and saliva during the WES indicated that the latter was at least largely uncontaminated. Rather, the novel SNVs that we identified, since they were mostly not proven to be cancerogenic, might be signs of clonal hemopoiesis rather than being associated with PMF development/progression [50].

In addition, since the patient population comprised elderly patients, we cannot exclude the fact that some mutations did just occur due to aging in hematopoietic stem cells. Further testing on younger patients (though it is not so common) would certainly help in clarifying this issue.

## 5. Conclusions

In conclusion, we found that most genetic lesions found in PMF patients share a myeloid and lymphoid lineage. This implies their occurrence in an early, uncommissioned hematopietic precursor, at least in a proportion of PMF cases, and provides a rationale for the possible occurrence of either myeloid or lymphoid blastic transformation. Further studies are recommended to analyze these results in a larger cohort of patients and to evaluate the role of distinct mutations in the pathogenesis of PMF and its transformation.

## Figures and Tables

**Figure 1 cancers-15-01785-f001:**
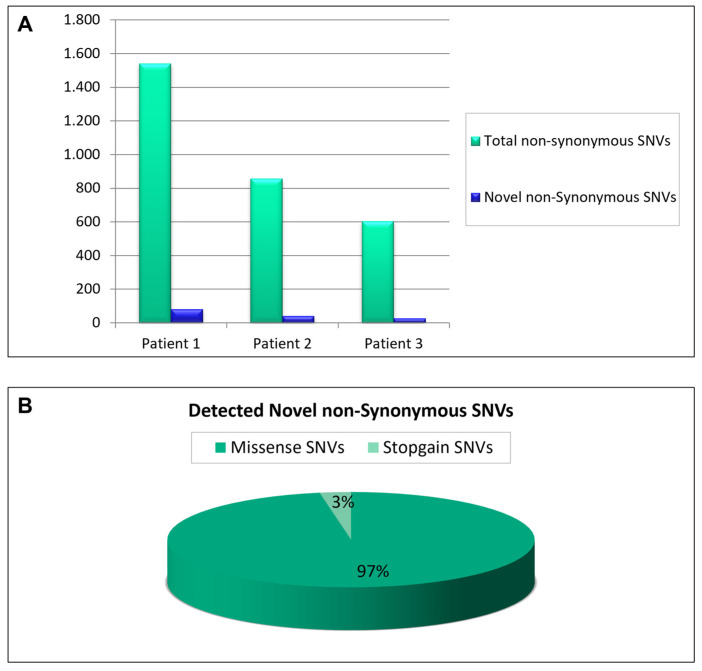
(**A**). Total and novel nonsynonymous single nucleotide variants. (**B**). Missense vs. stop/gain distribution of the novel SNVs.

**Figure 2 cancers-15-01785-f002:**
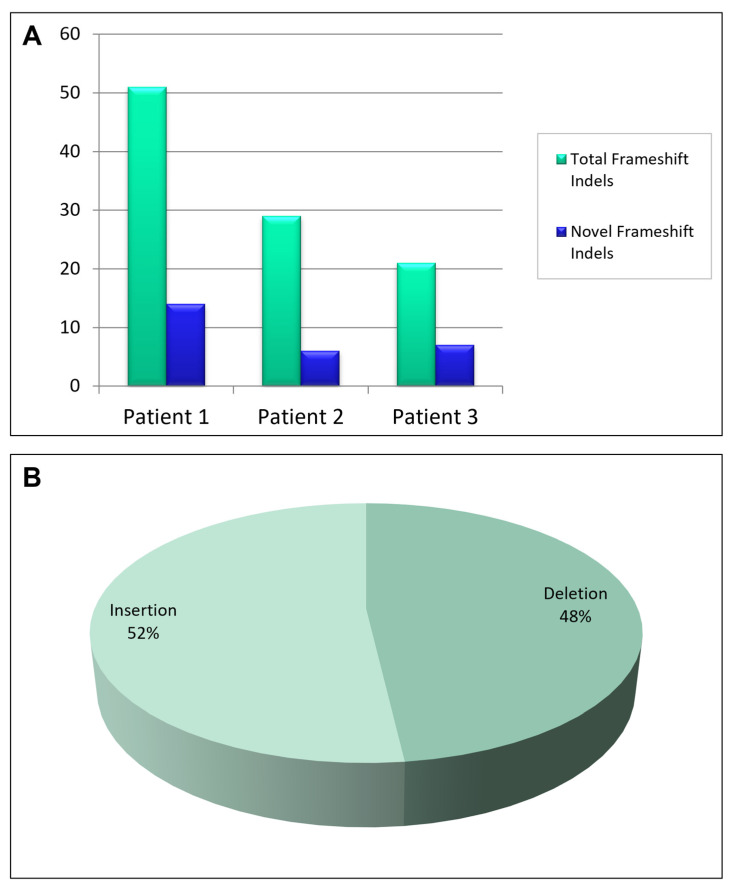
(**A**). Total and Novel frameshift inDels. (**B**). Distribution of Insertions vs. deletions. (**C**). Occurrence of gene involvement.

**Figure 3 cancers-15-01785-f003:**
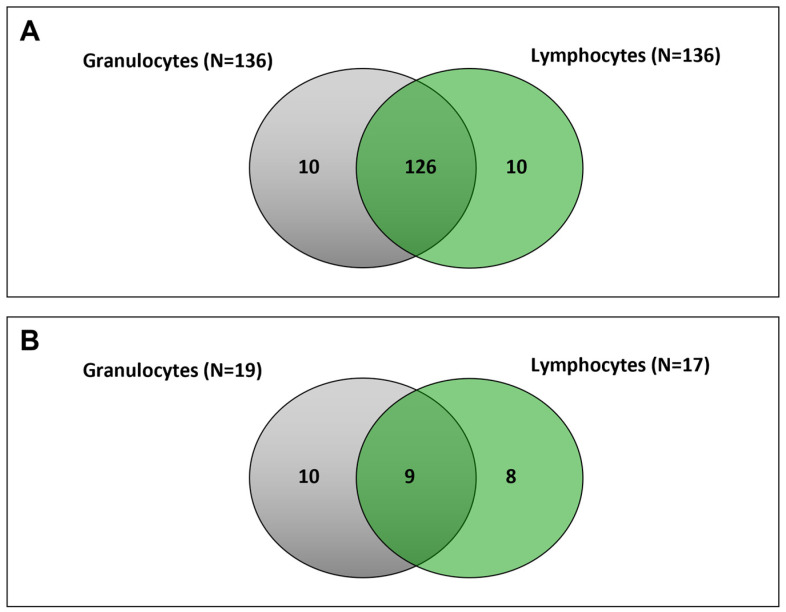
Comparison of SNVs occurring in either granulocytes or lymphocytes or both, overall (**A**) or limited to InDels (**B**).

**Table 1 cancers-15-01785-t001:** Patients’ characteristics.

Patient	Age (Years)	BM Fibrosis	Splenomegaly	Hb (g/dL)	WBC (×10^9^/L)	PLTs (×10^9^/L)
CC	72	I	Grade I	10.1	2.93	176
GT	80	III	Grade II	9.6	5.1	285
GB	73	I	Grade I	13.8	12.4	725

BM: bone marrow; Hb: hemoglobin; WBC: white blood cells; PLTs: platelets.

**Table 2 cancers-15-01785-t002:** Indels recorded at WES analysis.

SAMPLES	# TOTAL INDELS CALLED	# PASSED FILTER INDELS	# SOMATIC INDELS	# EXONIC/SPLICING INDELS *	# FRAMESHIT INDELS	# NOVEL INDELS *
GT_G_L	14.780	14.230	7.366	113	51	16 (o 14?)
GB_G_L	9.199	8.989	3.090	64	29	7 (o 6?)
CC_G_L	8.134	7.993	2.243	43	21	7

Indels recorded at WES analysis; G: granulocytes; L: lymphocytes; S: saliva. * NOT IN DBSNP AND 1000 GENOME.

## Data Availability

WES raw data are available at Alma Healthy Planet of Bologna University (https://centri.unibo.it/alma-healthy-planet/it (accessed on 7 March 2023)).

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
