# Peer review of "Use of Next Generation Sequencing to Define the Origin of Primary Myelofibrosis"

_cancers, 2023, doi:10.3390/cancers15061785_

Round 1

Reviewer 1 Report (Previous Reviewer 2)

The authors have responded to all my criticisms

Author Response

Thank you

Reviewer 2 Report (New Reviewer)

The authors employed molecular biology techniques (next-generation sequencing/WES) in their attempt to define the origin of primary myelofibrosis. The study is interesting and seems scientifically sound. However, the major drawback of the research is its small study sample -  samples from 3 PMF patients are not sufficient to draw accurate conclusions (to my view). In addition, all subjects were aged >70 years - how can we distinguish between molecular aberrations that occur due MPNs but also due to aging? I believe your research should have also included younger patients with PMF - maybe your effort can enable us to increase their life expectancy. In addition, it is not clear to me why some sections of the paper are written in red. Was this paper submitted to another journal and already peer-reviewed/rejected? If yes, it would be necessary to see the comments of the previous assessors and the letter of response to their concerns. The authors should discuss more about the potential applications of liquid biopsy and related techniques in BCR-ABL1-negative MPNs. In addition, what about the role of non-driver mutations? Genetic aberrations in included in the MIPSS70 (ASXL1EZH2IDH1/2SRSF2) or GIPSS (ASXL1, SRSF2U2AF1)? RUNX1? TP53? DNMT3A? IDH1/2? etc. Did these patients receive any treatment? JAK1/2 inhibitors? Hydroxycarbamide? Did treatment play any influence on your findings? Did you use samples collected at diagnosis prior to treatment initiation?  Genes should be written in italic. Reference 28 is crossed out in the reference list - please explain. Did you replace it? Authors contributions - please add the contribution of each author     

Author Response

Dear Referee,

we wish to thank you for yhe useful suggestions. We modified the text as recommended. Specifically, please find our point-by-point reply in the following:

  • The authors employed molecular biology techniques (next-generation sequencing/WES) in their attempt to define the origin of primary myelofibrosis. The study is interesting and seems scientifically sound. However, the major drawback of the research is its small study sample -  samples from 3 PMF patients are not sufficient to draw accurate conclusions (to my view). We have to agree with the referee. The study, however, confirmed important (and not yet confirmed data). We made it clear in the manuscript
  • In addition, all subjects were aged >70 years - how can we distinguish between molecular aberrations that occur due MPNs but also due to aging? I believe your research should have also included younger patients with PMF - maybe your effort can enable us to increase their life expectancy. This is quite an interesting point. We added it to discussion. We thank the referee for the advice (lane 374-76).
  • In addition, it is not clear to me why some sections of the paper are written in red. Was this paper submitted to another journal and already peer-reviewed/rejected? If yes, it would be necessary to see the comments of the previous assessors and the letter of response to their concerns. The manuscript was already revised. That’s correct. We submitted a point by point reply to the referees, as we addressed all their comments. The Editorial team should make them visible to the new referee as well.
  • The authors should discuss more about the potential applications of liquid biopsy and related techniques in BCR-ABL1-negative MPNs. We added this point to the discussion (lane 346-52)
  • In addition, what about the role of non-driver mutations? Genetic aberrations in included in the MIPSS70 (ASXL1EZH2IDH1/2SRSF2) or GIPSS (ASXL1, SRSF2U2AF1)? RUNX1? TP53? DNMT3A? IDH1/2? etc.We thank the referee for rising this relevant issue. Our patients were not assessed by NGS at diagnosis, this not allowing to classify them by the MIPSS70 score. The mutations affecting those genes and found by WES were described later. Due to the limited series, prognostic correlations were not possible of course.
  • Did these patients receive any treatment? JAK1/2 inhibitors? Hydroxycarbamide? Did treatment play any influence on your findings? Did you use samples collected at diagnosis prior to treatment initiation? Biological material was collected before treatment initiation. We added this specification to the Methods section (lane 122).
  • Genes should be written in italic. We revised the manuscript and we didn’t find inconsistencies in this regards. Genes are in capital italic and proteins just capital. Most likely, the format received by the referee did not allow a proper visualization.
  • Reference 28 is crossed out in the reference list - please explain. Did you replace it? This was a reply to a previous request from a referee. Now references are formatted in their final version.

Authors contributions - please add the contribution of each author . We apologize for the inconvenience. We realize that this was missing in this version. We ammended the text.   

We hope that you may find our manuscript now suitable for publication

Best regards

Round 2

Reviewer 2 Report (New Reviewer)

The authors have properly responded to my queries and the paper can be accepted for publication. 

This manuscript is a resubmission of an earlier submission. The following is a list of the peer review reports and author responses from that submission.

Round 1

Reviewer 1 Report

As correctly mentioned by the authors of this paper under review, the identification of the cells which are targeted by the driver mutations and which give rise to the Philadelphia-negative myeloproliferative neoplasms, including myelofibrosis, is a subject of great interest and of extensive investigations. By using a robust approach which included mutational landscaping by genome sequencing of CD34+ cells from more than 100 MPN and AML patients were published by the Levine laboratory. In this study, some of the MPN patients were analyzed both in the MPN and AML phase and included subjects with clonal hematopoiesis but no MPN/AML as control. The Levine laboratory has published the results of this study in Nature (2020 Nov;587(7834):477-4829 and rationally discussed them in a review in Blood (Blood 2020 Jul 2;136(1):61-70. doi: 10.1182/blood.2019000943).

 The paper by Visani et al addresses part of the questions discussed in the papers by Levine (mutational landascape at the myelofibrosis stage) by performing whole exome sequencing of myeloid and lymphoid cells purified from the peripheral blood from three patients with myelofibrosis. Cells collected from the saliva were used as non-hematopoietic controls. The experimental design of the study under review is far less robust in terms of experimental model and number of patients (only 3 patients) analyzed than the previous study from the Levine laboratory. The experimental model is less robust because the myeloid and lymphoid progeny of the CD34+ cells were analysed in bulk, saying nothing on the complexity of the active stem/progenitor cells at the single cell level. Still, the fact that the data presented in this paper support the conclusion obtained in the Levine laboratory is worthy to be published as confirmatory study.

The introduction and the discussion of the paper, however, must be completely rewritten to state of the art in the field and to indicate the caveats and limitation of the experimental model.

Two minor comments:

1)      The use of saliva as negative control may be confounded by the possible presence of neutrophils. Morphological analyses of all the cells used for DNA preparation should be presented to allow assessing possible cross-contaminations among samples.

2)      The figures are probably too many for a confirmatory study with only two patients. They should be reduced eliminating those that are speculative in nature. 

Reviewer 2 Report

The authors reported on NGS results in both myeloid and lymphoid circulating cells of patients with primary myelofibrosis with the aim to define the origin of the disease.  Whole exome sequencing revealed that the majority of the somatic mutations found in the two types of cells were in common. Particularly, they found 126/146 SNVs to be in common. The authors concluded that most genetic events likely to contribute to disease pathogenesis occurred in a non-commissioned precursor. They also found that only 9/27 InDels were in common: this result suggested that this type of lesion would rather contribute to disease progression, occurring at more differentiate stages.

Comment 1.

These results were derived from the analysis of three patients with primary myelofibrosis. What is said about the patients is that the pathological diagnosis was confirmed by at least 2 expert hematopathologists on bone marrow trephine biopsy. Patients were older than 72 years, in two of the cases BM fibrosis was grade 1, i.e. they were pre-fibrotic myelofibrosis. Two out of three cases had splenomegaly with variable degree of anemia, WBC and platelet count. No information about the IPSS risk score was reported.  No prior information about  the driver mutations (JAK2V617F, CALR, MPL) that are necessary for the diagnosis of the disease, or about the other frequent mutations necessary for the diagnosis of clonality  (according to WHO criteria) were provided. Only in the discussion of the paper, the authors reported  Indeed, we found JAK2V617F mutation in one of the samples which might seem relatively low, as the rate of this mutation in PMF is 50-60%, which could be explained by the low sample number  in our study. Curiously, neither CALR nor MPL mutations were observed as well in the initial cohort of 3 patients.”  No mention to the variant allele frequency was given. Thus, two of the three cases are triple negative cases that represent a rare variant of the disease (with many uncertainties about the true classification). Moreover, the clinical phenotype of the cases seem to be very particular: one pre-fibrotic patient has leucopenia and mild anemia, that is rare in such a variant. One with overt myelofibrosis and severe anemia has no splenomegaly. The authors should justify this selection of very particular cases and discuss whether the results obtained are transferable to the whole population of PMF patients.

 Comment 2

By whole exome sequencing the authors revealed novel as well as previously known SNVs and InDels. The number of mutations, the chromosomal  location and the number of mutations commonly found in granulocyte and lymphocyte were reported. However, the presence of driver mutations that are considered of pathogenetic  potential in myelofibrosis  (JAK2V617F, CALR, MPL, ASXL1, IDH1,IDH2, EZH2, SRSF2... ) was not reported. Moreover, the variant allele frequency of the mutations was never reported. Thus, the pathogenetic role of the mutations is not ascertained. The alternative hypothesis that the reported mutations reflect a condition of clonal hematopoiesis (CH) preceding the development of myelofibrosis should be discussed.  A number of papers have described CH in patients with solid cancers and hematological disorders. Lineage involvement of CH clones have evidenced  intercompartment heterogeneity for the mutations and found mutations of the granulocyte also in mature T- or B-cells (Hartmann L, et al. Compartment-specific mutational landscape of clonal hematopoiesis. Leukemia. 2022 Nov;36(11):2647-2655.)

Reviewer 3 Report

The authors address the question about the origin of primary myelofibrosis (PMF) using whole exom sequencing. It is a question of fundamental, biological interest.

Unfortunately, the authors attempt to reveal new pieces of information ends up in a rather descriptive study with vague conclusions. The manuscript would benefit from a more focused presentation and a deeper evaluation of the results.

Major points:

Page 2, line 44-46: The finding that one third of the indels were in common between myeloid and lymphoid cells does not by itself justify the conclusion that this type of genetic variants rather contributes to disease progression. Do they contribute to disease at all or are they just “passenger events”?

Page 3-4, line 90-105: To match the title and the abstract, the introduction should focus on giving a background to PMF as disease and what is known about its origin. The section here about vascular complications does not really contribute to the outlined topic or has a connection to the results, but rather makes the reader confused. It is also unclear what the next section listing some genes found in a specific study would add.

Page 4, line 106-108: The aim should be stated in a clearer way. “Tried to shed some light upon this matter” is not very informative. Moreover, the authors write that it should be done by “analyzing cytogenetic abnormalities” but the results focus more on SNVs and indels below the chromosomal level, although figures of chromosomal distribution are shown.

Page 4, section 2.1. Case selection: No statement about ethical approval or informed consent is made.

Page 4, table 1: How was this cohort selected? Was there a strategy in connection with driver mutation? At line 345-346 it is written “Curiously, neither CALR nor MPL mutations were observed as well in the initial cohort of 3 patients.”

Page 6-9, Results 3.1: The first section of the results concentrate on a more technical and general descriptive screening that do not answer the question about originating cell type. Table 2 shows general (technical) statistics about the sequencing and should rather be placed in a supplement. Table 3-4 and figure 1-2 show a total number of SNVs or indels respectively, divided between the patients, but not between granolucytes or lymphocytes. There is no deeper analysis about the findings, but there are some examples listed of genes associated with cancer together with references. This would rather be placed in the discussion, and instead focus on own findings in the results. There are also graphs showing chromosomal distribution, but there are not really any conclusions drawn from this.

Page 12, Results 3.3: Some of the findings were verified by Sanger sequencing but the rationale for choosing the selected genes it is not clear.

Minor points:

Page 3, line 73-81: Since this is a general background and not data from the current study, it is fine to state that JAK2 V617F occurs in 50-60% of cases. But to combine this with very precise % of the other mutation groups makes it a little strange, especially since it does not end up to 100%. It would be better ta write more general, e.g. CALR mutations are found in about one fourth of PMF cases.

Page 4, line 115-116: Nine patients used in validation cohort had been published previously and this paper is a reference for patients’ details. But there would have been helpful to the reader if details/parameters relevant for the current study have been presented here, perhaps in a supplement.

Page 10: Legend to figure 3D is missing.